# Identification of Novel Genes Associated with Partial Resistance to Aphanomyces Root Rot in Field Pea by BSR-Seq Analysis

**DOI:** 10.3390/ijms23179744

**Published:** 2022-08-28

**Authors:** Longfei Wu, Rudolph Fredua-Agyeman, Stephen E. Strelkov, Kan-Fa Chang, Sheau-Fang Hwang

**Affiliations:** Department of Agricultural, Food and Nutritional Science, University of Alberta, Edmonton, AB T6G 2P5, Canada

**Keywords:** root rot, *Aphanomyces euteiches*, bulk segregant RNA-seq analysis, differentially expressed genes (DEGs), field pea, SNPs

## Abstract

Aphanomyces root rot, caused by *Aphanomyces euteiches*, causes severe yield loss in field pea (*Pisum sativum*). The identification of a pea germplasm resistant to this disease is an important breeding objective. Polygenetic resistance has been reported in the field pea cultivar ‘00-2067’. To facilitate marker-assisted selection (MAS), bulked segregant RNA-seq (BSR-seq) analysis was conducted using an F_8_ RIL population derived from the cross of ‘Carman’ × ‘00-2067’. Root rot development was assessed under controlled conditions in replicated experiments. Resistant (R) and susceptible (S) bulks were constructed based on the root rot severity in a greenhouse study. The BSR-seq analysis of the R bulks generated 44,595,510~51,658,688 reads, of which the aligned sequences were linked to 44,757 genes in a reference genome. In total, 2356 differentially expressed genes were identified, of which 44 were used for gene annotation, including defense-related pathways (jasmonate, ethylene and salicylate) and the GO biological process. A total of 344.1 K SNPs were identified between the R and S bulks, of which 395 variants were located in 31 candidate genes. The identification of novel genes associated with partial resistance to Aphanomyces root rot in field pea by BSR-seq may facilitate efforts to improve management of this important disease.

## 1. Introduction

Field pea (*Pisum sativum* L.) is an economically important crop belonging to the Fabaceae family. It is characterized by high protein content in the seeds, ability to fix nitrogen and adaptation to cool seasons. These properties make field pea an ideal rotational crop in western Canada, where canola and wheat are the major crops. Canada is the largest field pea producer in the world (≈4.5 million tonnes in 2020), of which 82% is exported [1]. Unfortunately, the root rot complex, which involves several soilborne pathogens including *Fusarium* spp., *Aphanomyces euteiches*, *Pythium* spp., *Phytophthora* spp. and *Rhizoctonia* spp., is a major limitation to field pea production in Canada [2,3,4,5,6,7,8,9].

The oomycete *A. euteiches* is one of the most destructive pathogens of the root rot complex, infecting field pea at all stages of development. During the early stages of infection, *A. euteiches* causes damping-off and severe root rot. Later in the growing season, under favorable conditions, *A. euteiches* infection can destroy the root system completely, resulting in severe yield loss. Aphanomyces root rot (ARR) has been reported across the major pea-producing regions worldwide [10,11], including the Canadian provinces of Alberta and Saskatchewan [12,13]. Several studies have used molecular markers to evaluate the genetic structure of *A. euteiches* populations, with high genetic diversity reported in the USA [14,15] and low to moderate diversity identified in France [11,16]. Researchers have also examined the pathogenicity of *A. euteiches* by inoculation on differential pea cultivars, grouping isolates from France, Sweden, Denmark, the USA, Canada, Norway and New Zealand by race, virulence type or pathotype [11,14,16,17,18,19]. While some of these studies differed in the hosts tested and in some of their methodology, the highly virulent race/virulence type/pathotype I appears to be predominant in many pea-producing regions.

Cultural management practices and fungicidal seed treatments are insufficient to suppress ARR under field conditions [20,21,22,23], and genetic resistance may be the most promising tool to manage the disease. While field pea cultivars with complete resistance to ARR are not available, genotypes with partial polygenic resistance are used for disease management. Several plant introduction (PI) lines of pea have been developed to control ARR [24], and the pea cultivar ‘00-2067’ was reported to be tolerant to *A. euteiches* [22,25].

Previous studies have identified several genomic regions and quantitative trait loci (QTL) associated with partial resistance to ARR. Major QTL associated with resistance to ARR were reported on linkage group (LG) IV and VII, while minor QTL were reported on LG I, II, III and V [21,26,27,28,29]. These studies, however, used a limited number of PCR-based markers, resulting in low marker densities and relatively large QTL intervals. This makes it difficult to apply the flanking markers associated with the reported QTL for marker-assisted selection. Genotyping and mapping with high-density SNP arrays identified small-sized interval QTL associated with partial resistance to ARR in field pea [25,30].

Next-generation sequencing (NGS) is a revolutionary technology that has gained widespread use in crop improvement [31]. This technology can detect polymorphisms in DNA, mRNA and small RNA sequences and elucidate transcriptional processes, splicing patterns and gene expression levels [32,33,34,35,36,37]. RNA sequencing (RNA-seq) analysis has been applied in many crops, including maize [38], wheat [39,40,41], alfalfa [42,43] and soybean [44], to detect the presence and quantity of RNA under biotic and abiotic stress. Recent RNA-seq analyses of field pea have focused on the study of seed development [45], agronomic characters [46], root nodulation [47] and arbuscular mycorrhizal (AM) symbioses [48]. Bulked segregant RNA-sequencing (BSR-seq) technology combines NGS technology and bulked sergeant analysis [49] and has been used for the identification of gene-related markers associated with disease resistance in maize, wheat and canola [49,50,51,52]. Liu et al. [50] detected novel polymorphic markers associated with the ‘glossy’ (gl3) phenotype of maize in a small-sized interval, leading to the cloning of this gene. BSR-seq was also used for molecular characterization of the resistance genes *Yr15*, *YrZH22, YrMM58*, *YrHY1*, *Yr26*, *Pm4b* and *PmSGD* in wheat [52], and to identify the clubroot resistance gene *Rcr1* in canola and for the identification of markers for marker-assisted selection [51].

At present, an increasing number of pathway databases are available for exploration of visualized biological mechanisms with associated open reading frames (ORFs), genes and proteins, including the Kyoto Encyclopedia of Genes and Genomes (KEGG) [53], Plant Reactome [54], MetaCyc [55] and others. Many legume crops, not including field pea, are available in the KEGG database, which is usually used for annotation of pea nucleotide sequences against other available legume crops, such as chickpea, soybean and *Medicago truncatula* [46,47]. In addition to biological pathway databases, the gene ontology (GO) consortium has been developed to help evaluate the roles of genes and gene products [56]. The GO terms contain three components: cellular components, molecular functions and biological processes, of which GO biological processes are similar to the KEGG pathway, but focus on the molecular events of a gene, rather than a gene network [57]. Currently, sequence blast, biological pathway and GO annotation are available for field pea in the Pulse Crop Database (www.pulsedb.org/; accessed on 1 December 2021).

Plant defense mechanisms associated with the interaction between field pea and *A. euteiches* are still not clear. Generally, plants initiate pattern-triggered immunity (PTI) by recognizing pathogen-associated molecular patterns (PAMPs) [58]. When pathogens produce effectors to suppress PTI, plants can recognize the special effectors to activate effector-triggered immunity (ETI) [59]. Jasmonic acid (JA), ethylene (ET) and salicylic acid (SA) play important roles in the plant immune response, with the genes controlling these signaling pathways often evaluated in studies of plant defense mechanisms [60,61,62]. In addition, abscisic acid, auxin, brassinosteroids and gibberellins can also be involved in plant defense signaling [63].

Molecular studies of field pea have lagged behind other pulses due to its large genome size (4.45 Gb; 2 *n* = 14) and the highly repetitive nature of the genome [64,65]. In pea, RNA-seq has been used only to evaluate transcriptional gene expression levels during the interaction between field pea and *Rhizobium* [46,47]. Some de novo assembly studies also used RNA-seq analysis to evaluate the transcriptome of pea seed development [45,66]. Disease-related markers in genomic regions associated with resistance to ARR are essential for marker-assisted selection. The objectives of this study were to: (1) confirm the candidate interval for resistance to ARR through BSR-seq analysis, (2) develop SNP markers to fine map the QTL associated with root rot resistance in ‘00-2067’ and (3) identify differentially expressed genes and predict the pathway(s) associated with resistance to *A. euteiches*.

## 2. Results

### 2.1. Root Rot Severity and Growth Parameters

ANOVA indicated a significant genotypic effect on ARR severity, vigor, plant height and dry foliar weight, suggesting that a high portion of heritable variance was transmitted from the parental cultivar to the RIL population (Table 1). Significant differences between the parental cultivars ‘Carman’ and ‘00-2067’ were detected for all traits except dry foliar weight, with estimated means and stand error (SE) of 6.72 ± 1.9 and 2.2 ± 1.3 for disease severity, 1.7 ± 1.1 and 3.3 ± 0.6 for vigor, 8.0 cm ± 4.8 cm and 19.7 cm ± 5.1 cm for height, and 1.5 g ± 0.7 g and 1.3 g ± 0.6 g for dry foliar weight, respectively. Disease severity was negatively correlated with plant height (−0.59 < r < -0.22), vigor (−0.98 < r < −0.89) and dry foliar weight (−0.80 < r < −0.14), which indicated the adverse impact of ARR on overall plant growth. Due to the significance among three replications for root rot severity, plant height, vigor and foliar weight, the correlation coefficients were analyzed to indicate the coincidence among three replications for all the traits. High correlation coefficients among the means from three greenhouse studies were found for disease severity (0.51 < r < 0.58, *p* < 0.001), vigor (0.48 < r < 0.60, *p* < 0.001), plant height (0.72 < r < 0.82, *p* < 0.001) and dry foliar weight (0.42 < r < 0.72, *p* < 0.001), illustrating the stable reaction of the RIL population to ARR (Figure 1). The individuals used to generate the bulks were selected based on extreme scores for disease severity. The highly resistant lines (DS < 2.5) constituted 33% of the total RIL population, while the highly susceptible lines (DS > 5.5) represented 22% of the population.

### 2.2. RNA-Seq Analysis and Sequence Alignment

The RNA-seq analysis generated 44,595,510–51,658,688 and 43,848,192–47,866,574 raw read pairs for the three replicates of R and S bulks, respectively. The Q ≥ 30 values ranged from 93.0% to 93.9%, which suggested high quality and accurate sequencing data. In addition, 98.1–99.4% of the reads for the R bulks were aligned to the field pea reference genome, Pisum_sativum_v1a.fa (https://urgi.versailles.inra.fr/download/pea/Pisum_sativum_v1a.fa; accessed on 26 April 2021), compared with 99.0–99.5% of the reads for the S bulks. Furthermore, 83.4–85.1% of the reads for the R and S bulks were exonic, which indicated that high portions of the tested sequences were located in the gene-encoding region. The expression level of 44,756 genes was evaluated, 56.8–57.4% of which were expressed in the R bulks and 56.8–57.3% of which were expressed in the S bulks.

### 2.3. Selection of Differentially Expressed Genes

With a threshold of |log2 FC| > 2, three single R-S pairs selected 601, 1416 and 977 DEGs for R1-S1, R2-S2 and R3-S3, respectively. By taking advantage of the DESeq analysis using the Wald test, significances were detected in 44 and 21 DEGs for the two in silico mixes of (R1 + R3) vs. (S2 + S3) (Figure 2A) and (R1 + R2 + R3) vs. (S1 + S2 + S3) (Figure 2B), respectively. Therefore, DESeq analysis determined 46 DEGs, of which 25 DEGs were down-regulated and 21 were up-regulated in the R mixed bulks compared with the S mixed bulks (Figure 2 and Figure 3A). A total of 2726 DEGs were identified by the R and S bulks comparison using either single R-S pairs or two in silico mixed bulks, which were located on the seven pea chromosomes: chr1LG6 (316), chr2LG1 (230), chr3LG5 (304), chr4LG4 (332), chr5LG3 (423), chr6LG2 (368) and chr7LG7 (383) (Figure 3A). A total of 1020 DEGs were found unique to a single method, while 1706 DEGs were identified by more than one method (Figure 3B).

### 2.4. Identification of Variants between the R and S Bulks

Frequent variants were identified for the six samples, of which the R bulks contained 238.9–254.9 K SNPs, while the S bulks contained 234.4–265.6 K SNPs. Biallelic unique SNPs detected in the R bulks consisted of 89.6–89.7% (214.4–228.5 K) of the total SNPs. A similar percentage (89.5–89.6%; 209.9–238.0 K) of the SNPs in the S bulks was biallelic unique. The polymorphic SNPs were selected for three individual R-S bulk pairs, numbering 14.9 K (R1 vs. S1), 14.6 K (R2 vs. S2) and 15.6 K (R3 vs. S3). For the two in silico mixes, the numbers of common SNPs within the R bulk were 160.3 K (R1 + R3) and 138.3 K (R1 + R2 + R3), while the numbers for the S bulk mixes were 151.9 K (S2 + S3) and 136.9 K. For the bulk mixes with two clustered replicates, the comparison of common SNPs between the R and S bulks identified 120.9 K (63.2%) monomorphic SNPs and 70.4 K (36.8%) polymorphic SNPs. For the bulk mixes with all three replicates, monomorphic and polymorphic SNPs were 107.7 K (64.3%) and 59.7 K (35.7%). Overall, 344.1 K polymorphic SNPs were identified based on the R and S comparison of three single R-S bulk pair and two in silico mixes, of which 296.6 K were aligned to seven chromosomes of field pea. The SNP densities of each chromosome were 103.7 SNPs/Mb on chromosome 1 (LGVI), 104.3 SNPs/Mb on chromosome 2 (LGI), 98.1 SNPs/Mb on chromosome 3 (LGV), 120.2 SNPs/Mb on chromosome 4 (LGIV), 10.9 SNPs/Mb on chromosome 5 (LGIII), 96.5 SNPs/Mb on chromosome 6 (LGII) and 123.9 SNPs/MB on chromosome 7 (LGVII). The most frequent variant regions were centered on the top and middle of chromosome 2 (LGI), bottom of chromosome 3 (LGV), middle of chromosome 5 (LGIV), top of chromosome 6 (LGII) and bottom of chromosome 7 (LGVII) (Figure 4).

### 2.5. Functional Enrichment Analyses of Differentially Expressed Genes

The 2356 selected DEGs that were aligned on seven pea chromosomes were used to search GO terms associated with disease response and root growth, as well as pathways related to JA, ET and SA signaling in the Pulse Crop Database (Figure 5 and Appendix A). Thirty DEGs were linked to the GO biological process associated with the plant defense response, including GO: 0006952, GO: 0031347, GO: 0031348 and GO: 0031349. Meanwhile, three DEGs were associated with the plant immune response (GO: 0006955), which were coincidently present in the defense-response-related DEGS. In addition, three DEGs were annotated to the GO biological process of root development, including GO: 0010015, GO: 0010053, GO: 0022622 and GO: 0048364. For those DEGs related to the plant defense pathway, eight, one and two DEGs were involved in jasmonic acid biosynthesis, ethylene biosynthesis I and methyl-salicylate metabolism, respectively. A BlastN search of the Pulse Crop Database indicated that the 30 defense-response-related DEGs were associated with molecular functions including protein binding, ADP binding, abscisic acid binding, protein phosphatase inhibitor activity and signaling receptor activity. The DEGs Psat1g156800, Psat1g156920, Psat1g157160 Psat2g013520, Psat3g126600 and Psat4g025040 were related to the biological process of signaling defense response. All eight DEGs linked to jasmonic acid biosynthesis were annotated to the oxidation-reduction process. Jasmonic acid biosynthesis was not only related to signals that stimulated plant defenses against pathogens, herbivory, wounding and abiotic stress, but also controlled plant developmental processes such as root elongation. Both DEGs related to methyl-salicylate metabolism were associated with hydrolase activity.

### 2.6. Analysis of Differential Expressed Genes and SNPs in the Target Region

The assessment of polymorphisms in the R and S bulks identified a range of variants among the 44 selected DEGs on 7 pea chromosomes (Appendix A). A total of 395 SNPs were detected within 31 annotated DEGs. In contrast, no SNPs were detected for 13 DEGs, including Psat1g110880, Psat1g156920, Psat4g087360, Psat4g201600, Psat5g066680, Psat5g242440, Psat5g242600, Psat5g289880, Psat5g291280, Psat5g291320, Psat6g011200, Psat6g098320 and Psat6g164080. Psat3g074240 contained the most (50) variants and a density of 15.2 SNPs/Kb, while Psat7g067840 included 11 SNPs but showed the highest SNP density (20.2 SNPs/Kb).

In a previous study [19], we found that the major QTL associated with partial resistance to *A. euteiches* in the pea ‘00-2067’ was located on chromosome 4 (LG IV), while several minor to moderate effect QTLs were located on chromosomes 5 (LG III), 6 (LG II) and 7 (LG VII). In the current study, 10 of the 44 annotated DEGs were located in genomic regions reported by Wu et al. [19]. Eight of the DEGs, Psat4g152600, Psat4g180200, Psat4g180800, Psat4g184760, Psat4g185080, Psat4g186560, Psat4g201520, Psat4g201600, were located in the most stable genomic regions, *AeMRDC1-Ps4.1* and *AeMRDC1-Ps4.2*, reported to be associated with ARR resistance [19]. In contrast, Psat3g069000 and Psat3g074240 were located in the minor effect QTL *Hgt-Ps5.1*. The polymorphic SNP marker PsCam027331_15987_254 in the genetic map constructed by Wu et al. [19] was annotated to Psat4g186560. In this study, 115 SNPs were found within the 8 DEGs on chromosome 4, ranging from 0 to 46. For the DEGs in *Hgt-Ps5.1*, 14 and 50 SNPs were detected within Psat3g069000 and Psat3g074240, respectively. These SNPs provided a promising source of markers to merge the gap in the previous genetic map. The remaining 34 DEGs were not reported in the previous study, with 216 SNPs detected in 22 novel genes on the 7 pea chromosomes.

The comparison of the R and S bulks in this study identified 250 polymorphic SNPs on all 7 pea chromosomes in the DEG regions related to the defense response, of which 240, 21, 11 and 10 SNPs, respectively, were linked to GO:0006952 (29 DEGs), GO:0006952 (4 DEGs) and GO:0031348 (3 DEGs). Only one of three DEGs associated with root system development, Psat5g007800, contained ten SNPs. In the case of the signaling defense response involving DEGs on chromosomes 2, 3, 4 and 6, a total of 45 polymorphic SNPs were detected in Psat2g149200 (13 SNPs), Psat3g069000 (14 SNPs), Psat4g184760 (46 SNPs) and Psat4g185080 (22 SNPs). However, there were no polymorphic SNPs in the other four DEGs on chromosomes 5 and 6. In Psat1g105280 and Psat3g026920, 16 and 24 SNPs were identified, respectively, which participated in the SA signaling pathway. There was no polymorphic SNP relating to the ethylene signaling pathway.

## 3. Discussion

Aphanomyces root rot caused by *A. euteiches* is a major limitation to field pea production and has attracted significant attention from researchers in recent years. The use of partially resistant cultivars is the most effective method to control this disease, particularly given the lack of fully resistant genotypes. The pea cultivar ‘00-2067’ was reported to be partially resistant to infection by *A. euteiches* under field conditions [22], and this partial resistance to ARR as well as to Fusarium root rot was further explored in recent studies [25,67]. The most stable and major QTL for resistance to the root rot complex were mapped to two genomic regions on chromosome 4, while minor to moderate QTL were located on chromosomes 5, 6 and 7 [19,62].

The QTL identified by Wu et al. [25,67] were determined using an F_8_ RIL population derived from the cross ‘00-2067’ (root rot-resistant parent) × ‘Reward’ (susceptible parent). To study the inheritance of the identified QTLs in a different genetic background, we crossed the ARR and Fusarium root rot resistant parent ‘00-2067’ with the susceptible cultivar ‘Carman’ and developed RIL of 135 individuals. The results of the current study validated the stability of genetic resistance in ‘00-2067’. Similar to our previous studies [25,67], significant genotypic effects, a high correlation coefficient within each trait and a negative correlation of root rot severity with vigor and plant height were observed in the RIL population derived from ‘00-2067’ × ‘Carman’. The frequency distribution of the disease severity data for the RIL population suggested that the resistance in ‘00-2067’ was transferred to the progenies in the RIL lines. This confirms the potential of ‘00-2067’ as a resistance source for pea breeding programs focused on root rot diseases. Several studies have evaluated the polygenetic resistance to ARR in field pea. The QTL associated with partial resistance to *A. euteiches* were identified using PCR-based markers. However, the limited number of markers, low marker density and lack of background gene information make it difficult to apply the identified markers for use in MAS [21,26,27,28,29]. Next-generation sequencing technology has accelerated the development of many SNPs and other markers based on gene-encoding sequences in the field pea genome [68]. These large numbers of markers can facilitate fine mapping of QTL and, more importantly, the candidate genes associated with resistance to *A. euteiches* [25,30]. Tayeh et al. [63] developed a pea SNP array based on agronomic traits. Genetic studies by Desgroux et al. [30] and Wu et al. [25] used the pea SNP array to identify QTLs associated with ARR.

RNA-seq technologies could provide a deeper understanding of gene function, regulatory networks and the associated biological processes and pathways of the target traits, such as agronomic characters and plant defense mechanisms [47,48,69]. The information at the genome and transcriptome levels can be used to detect novel genes related to the target traits. Indeed, RNA-seq analysis has been applied to characterize the transcriptomes of several major crop species, including maize, wheat and soybean [38,44,69,70]. Transcriptomic evaluations of field pea based on RNA-seq analysis have revealed genes associated with biological processes such as nodulation, nitrogen fixation and the plant immune response [47,48,62].

Study of the field pea genome has lagged behind that of other legumes due to its large size and complexity. As such, gene function studies in field pea were usually obtained by comparison with *Medicago truncatula* (barrel clover), *Cicer arietinum* (chickpea), *Glycine max* (soybean) and *Arabidopsis thaliana* [47,48,62]. The pea reference genome was first published in 2019, along with abundant gene function and metabolism pathway information [65].

BSR-seq analysis, which can rapidly and economically detect target traits, is an improvement over RNA-seq that has been applied in maize and wheat [49,50]. To our knowledge, this is the first study of its kind where BSR-seq has been used to detect novel genes in field pea. In this study, 4.3–5.1 Gb read pairs obtained from all R and S bulks were used for genome assembly, of which 98.1–99.5% were aligned to the reference genome, with about 84% of the reads located in exonic regions of the pea genome. Therefore, the results are comparable to those reported by Kreplak et al. [65] for the pea reference genome. Sudheesh et al. [46] applied de novo assembly and identified around 140 K contigs in field pea. However, they reported that only 50% of contigs were annotated, mostly to *M. truncatula* and soybean. Only 3.3% of the contigs matched to the gene-encoding sequences in pea. Malovichko et al. [66] also conducted de novo assembly without the pea reference genome, producing 25,756 contigs distributed between 2112 genes, much less than the 44,756 genes evaluated in the current study. The improvement of sequence matching in this study largely reflected the availability of the pea reference genome.

Eight of the DEGs associated with jasmonic acid biosynthesis in this study have been reported to be involved in oxidation–reduction processes, which can play important roles in the plant immune response [71,72,73]. Seven of the eight DEGs involved in oxidation–reduction, with the exception of Psat6g098320, have been reported to control the reaction: linolenate + oxygen → 13(S)-HPOTE, which is a key step in jasmonic acid biosynthesis. Jasmonic acid plays an essential role in plant growth and development, as well as in the plant immune response [74,75]. Two DEGs involved in methyl salicylate metabolism were both annotated to the GO gene function of hydrolase activity, which can play an important role in plant defense responses by regulating ADP-ribose and NADH [76]. Only one DEG was linked to ethylene biosynthesis. Ethylene biosynthesis is essential in regulation of lesion mimic mutant vad1-1, which is related to propagative hypersensitiveness [77].

Eight selected genes were mapped to the most stable QTL region, *AeMRDC1-Ps4.1* and *AeMRDC1-Ps4.2* associated with partial resistance to ARR, while two genes mapped to the minor QTL *Hgt-Ps5.1* [25]. The 34 remaining genes identified in this study were novel and mapped to chromosomes 1, 2, 3, 4, 5 6 and 7. In a previous study using a 13.2K SNP array, we mapped the major QTL for partial resistance to ARR to chromosome 4 and minor-moderate QTL to chromosomes 5, 6 and 7 [19]. The present BSR-seq analysis led to the identification of novel genes associated with partial resistance to ARR in the pea ‘00-2067’, complementing the earlier work. While QTL analysis can successfully connect phenotypic traits with their genetic component, the involvement of multiple genes with small effects, QTL interactions and large variations in different environments can make it difficult to apply this approach in breeding programs [78,79]. As such, the evaluation of associated gene functions and pathways is essential to fill the gaps between QTL analysis and its application in a breeding program. Derakhshani et al. [80] combined QTL mapping with RNA-seq analysis not only to identify candidate genes associated with cadmium tolerance in barley, but also to evaluate the effect of single QTL by RNA-seq analysis. Similarly, a combination of QTL mapping and RNA-seq analysis revealed candidate genes in the QTL and potential mechanisms of salt-stress tolerance in rice [81].

Quillévéré-Hamard et al. [82] evaluated several NILs carrying QTL previously found [29] to contribute to resistance to ARR under both field and greenhouse conditions and reported a high potential for the use of these QTL in gene pyramiding and MAS breeding. In the current study, 344.1 K SNPs were identified to be polymorphic between the R and S bulks. The mean SNP density on the seven pea chromosomes ranged from 96.5 to 120.2 SNPs/Mb, except for chromosome 5, which had a very low density of 10.9 SNPs/Mb (Figure 4). The low variant density on this chromosome suggests that this genomic region was conserved in the population used in this study. Large intense regions of polymorphic variants with SNP densities greater than 500 SNPs/1 Mb were found on chromosomes 2, 4 and 7, along with narrow intense regions on the bottom of chromosome 3 and top of chromosome 5 (Figure 4). This suggests that the SNPs identified in this study could be valuable in the detection of novel QTL in ‘00-2067’, controlling resistance to ARR.

## 4. Materials and Methods

### 4.1. Plant Material

The semi-leafless parental cultivar ‘00-2067’, derived from the crosses (PH14-119×DL-1)7 × (B563-429-2 × PI 257593) × DSP-TAC, produces white flowers and a wrinkled seed coat, and was reported to be tolerant to ARR and *Fusarium* spp. [22,25,68]. The pedigree of the susceptible cultivar ‘Carman’, which produces white flowers and green cotyledons, is unknown. An F_8_ RIL population was generated by single-seed descent (SSD) from the parents ‘Reward’ and ‘00-2067’ and was comprised of 135 individuals.

### 4.2. Root Rot Assessment

Greenhouse studies were carried out in a randomized complete block design with 12 replicates. Seeds of the RIL were sterilized in 1% NaClO for 1 min and washed three times in sterilized water. Four seeds of each RIL were germinated on moistened filter paper in a Petri dish and then transplanted into 7 cm × 7 cm × 10 cm plastic pots containing sterilized nutrient soil mixture (Cell-TechTM, Monsanto, Winnipeg, MB, Canada). Isolate Ae-MRDC1 of *A. euteiches*, which had been previously collected from a disease nursery in Manitoba, Canada and classified as pathotype I [25], was used for all inoculations. An oospore suspension was produced following Wu et al. [19] and adjusted to a final concentration of 1 × 10^5^ oospores mL^−1^. Before the rootlets of the transplanted seedlings were covered with soil mixture, each seedling was inoculated with a 1 mL aliquot of the spore suspension. The plants were maintained under a 12-h photoperiod with day temperatures of 22–28 °C and night temperatures of 15–18 °C. Plant height, dry foliar weight, vigor (0–4) and disease severity (DS) (0–9) were evaluated after 3 weeks according to Wu et al. [25,83]. Briefly, the root rot severity ratings were: 0, healthy seedling; 1–2, slight necrosis; 3–4, moderate necrosis; 5–6, extensive necrosis and 7–9, root system and stem severely damaged, seedling completely dead. Thus, RILs with a DS < 2.5 and DS > 5.5 were regarded as resistant and susceptible, respectively, to obtain sufficient individuals (~20% of total RILs) for subsequent bulk construction. The greenhouse studies were repeated three times.

### 4.3. Bulks Construction and RNA Extraction

Resistant (R) and susceptible (S) bulks were generated from RILs exhibiting extreme and stable disease reactions to *A. euteiches*. Twenty-five RILs with stable resistance (DS < 2.5) to *A. euteiches* and twenty-five RILs with stable susceptibility (DS > 5.5) were selected to form the R and S bulks, respectively. About 1 cm of the main root tissue from each of the 25 individuals from each bulk was excised and mixed for RNA extraction. Each bulk contained three biological replicates. The mixed root tissues of each replicate from each bulk were ground into a powder in liquid nitrogen; the RNA was extracted from the powdered root tissue as described by Zhou et al. [84]. Briefly, 0.1 mL root powder was homogenized in 1 mL Trizol (Ambion-Life Technologies, Carlsbad, CA, USA) for 15 min, treated with 0.2 mL chloroform (Fisher Chemical, Fair Lawn, NJ, USA) for 10 min and precipitated using 0.5 mL 2-propanol (Fisher Chemical, Fair Lawn, NJ, USA) for 3 h. The extracted RNA was cleaned using an RNeasy Mini Kit (Qiagen, Hilden, Germany), and the DNA component of the RNA sample was eliminated by treating it with DNAse (Qiagen, Hilden, Germany) for 15 min at room temperature. The RNA concentration of each sample was measured in a NanoDrop 2000c Spectrophotometer (Thermo Fisher Scientific, Waltham, MA, USA) and adjusted to 50 ng/μL. An Agilent 2200 TapeStation system (Agilent, Santa Clara, CA, USA) was used to confirm the quality and purity of each RNA sample.

### 4.4. RNA-Seq and Sequence Alignment

The cDNA library was prepared using an Illumina TruSeq stranded mRNA kit (Illumina; San Diego, CA, USA) and sequenced with a NovaSeq (Illumina). Sequence alignments were performed using a STAR (v2.7.3a) aligner, and the paired-end reads were aligned to the reference *P. sativum* (ea) genome downloaded from: https://urgi.versailles.inra.fr/download/pea/Pisum_sativum_v1a.fa; accessed on 26 April 2021. The generic feature format (GFF) file was downloaded from: https://urgi.versailles.inra.fr/download/pea/Pisum_sativum_v1a_genes.gff3; accessed on 26 April 2021. Reads that mapped to ribosomal RNA and the mitochondrial genome were removed before performing alignment. The raw read counts were estimated using HTSeq v. 0.11.2. Read counts were normalized using the package ‘DESeq2’ [85], and hierarchical clustering analysis was performed for the normalized counts. Euclidean distance and the complete linkage clustering method were used for hierarchical clustering. Analysis was performed using R v. 3.5.2 and the additional packages: ggplot2, reshape2 and ggrepel.

### 4.5. Identification of Variants between R and S Bulks

Genohub Inc. (Austin, TX, USA) generated VCF files to capture the variants for each sample of R and S bulks. The SNP and biallelic SNP numbers were obtained using the package ‘SNPRelate’ [86] in R v. 3.5.2. The variants found were then used to identify SNPs with the R package ‘pegas’ [87]. To improve the accuracy of statistics, the bulk comparisons for polymorphic SNP were conducted using three single R-S pairs as described by Yu et al. [51] (R1-S1, R2-S2 and R3-S3), as well as two in silico mixes as described by Ramirez-Gonzalez et al. [77], including (i) two clustered S bulks (S2 + S3) and two clustered R bulks (R1 + R3) based on principal component analysis (PCA) and hierarchical clustering analysis (Appendix A) and (ii) all the susceptible (S1 + S2 + S3) and resistant bulks (R1 + R2 + R3). The common variants within biological replicates in the R and S bulks for the two in silico mixes, respectively, were detected with the R packages ‘RCurl’ [88], ‘purrr’ [89], ‘VariantAnnotation’ [90], ‘GenomicRanges’ [91] and ‘Rsubread’ [92]. Comparisons of common variants between the R and S bulks for the three single bulked pairs and two in silico mixes were conducted using the R package ‘plyr’ [93] to obtain monomorphic and polymorphic SNPs. The polymorphic SNP density distribution was generated with the R package ‘rMVP’ [94].

### 4.6. Disease-Related Gene Expression Analysis

The aligned reads were used for the estimation of the expressed genes. The raw read counts were estimated using HTSeq (v0.11.2). The script HTseq-count is a tool for RNA-seq data analysis. The SAM/BAM file and a GTF or GFF file with gene models were used to count the number of aligned reads for each gene that overlapped with exons. Only reads that mapped unambiguously to a single gene were counted, whereas reads that aligned to multiple positions or overlapped with more than one gene were discarded. Read count data were normalized using DESeq2. Additionally, the expression of the aligned reads was estimated using cufflinks v. 2.2.1. The expression values were reported in fragments per kilobase per million (FPKM) for each gene. The significance of differentially expressed genes (DEGs) between the R and S bulks was determined based on the log2 fold change (|log2 FC| > 2) for the three single bulk pairs. To better account for the variants component of the two in silico mixes, DEGs were also selected using the package ‘DESeq2’ in R v. 3.5.2 (P_adj_ < 0.05). The accessions of selected genes were used in searches with BlastN and to search gene ontology (GO) terms and pathways in the Pulse Crop Database (www.pulsedb.org/; accessed on 1 December 2021) to determine gene function and biological processes, as well as associated pathways involved in the disease response.

## 5. Conclusions

BSR-seq analysis was used to identify SNPs in field pea at the gene expression level. The variant annotation contributed to the development of SNPs for detecting novel QTL controlling partial resistance to ARR in field pea. The jasmonic acid, ethylene and salicylic acid-related pathways as well as the GO biological process associated with the defense response, immune response and root development detected in this study may be important for understanding the field pea/*A. euteiches* interaction. The results indicate that the pea cultivar ‘00-2067’ can be used in resistance breeding programs and for the development of markers for use in MAS.

## Figures and Tables

**Figure 1 ijms-23-09744-f001:**
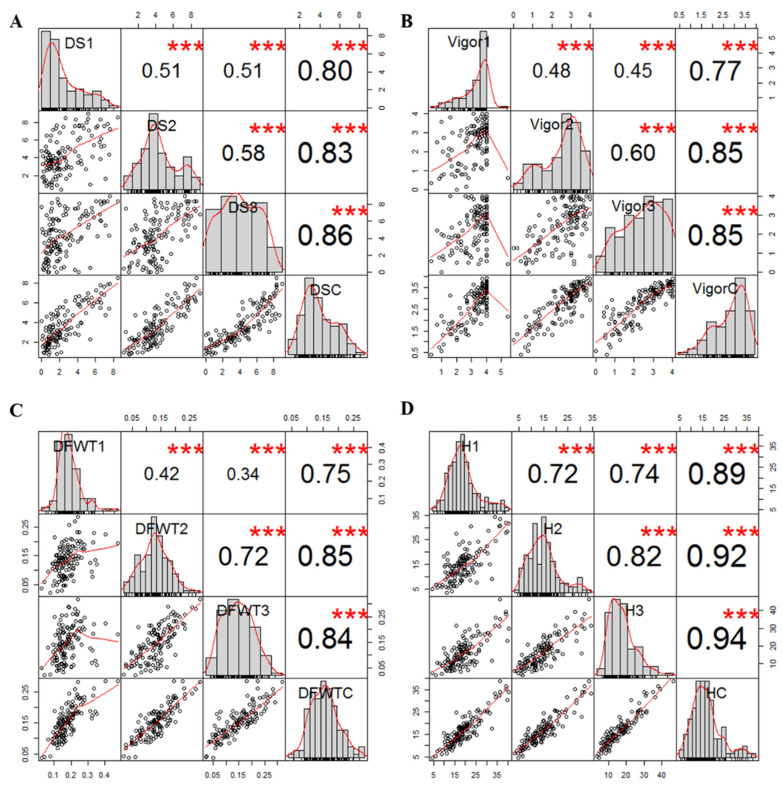
Correlation analysis of estimated mean of three single greenhouse experiments and combined total data for (**A**) root rot severity, (**B**) vigor, (**C**) dry foliar weight and (**D**) height of pea inoculated with *Aphanomyces euteiches*, illustrating the significant correlation among all variables for each trait. The bar graphs indicate the frequency distributions across the diagonal. The correlation coefficients with a significance level (*** indicates *p* < 0.001) and scatter plots between pairs are shown above and below the diagonal, respectively.

**Figure 2 ijms-23-09744-f002:**
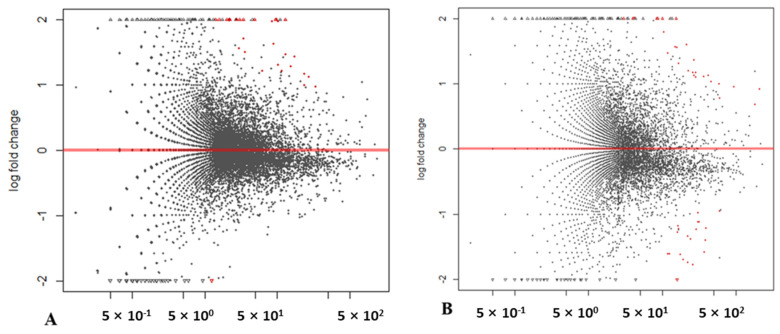
MA-plot from base means (*x*-axis; ‘M’) and the average of log fold changes (*y*-axis; ‘A’), indicating differentially expressed genes in pea resistant (R) or susceptible (S) to Aphanomyces root rot in DESeq analyses of (**A**) an in silico mix with three replicates in resistant (R) and susceptible (S) bulks, and (**B**) an in silico mix with two replicates in R and S bulks. Red spots indicate genes with P_adj_ < 0.05.

**Figure 3 ijms-23-09744-f003:**
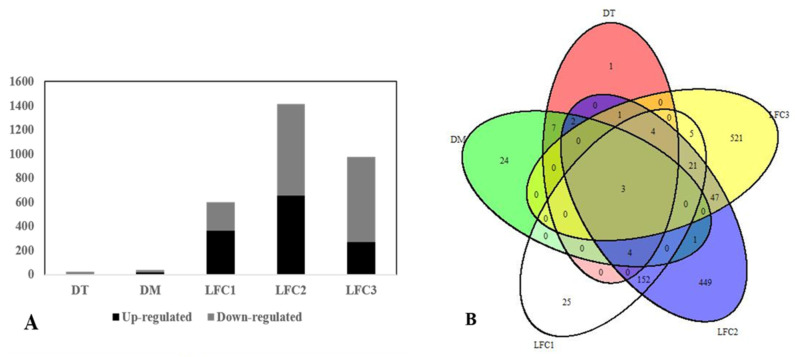
Number of differentially expressed genes (DEGs) in pea resistant (R) or susceptible (S) to Aphanomyces root rot, as detected by two in silico mixes and log2 fold change comparisons, as well as the overlap among these DEGs. (**A**) Overview of the number of significantly up-regulated and down-regulated genes. (**B**) The overlap in DEGs in a Venn diagram. ‘DT’ indicates DEGs determined by DESeq analysis from an in silico mix with three replicates; ‘DM’ indicates DEGs determined by DESeq analysis from an in silico mix with two replicates and ‘LFC1-3’ indicates log2 fold change comparison of individual R-S pairs (R1-S1, R2-S2 and R3-S3).

**Figure 4 ijms-23-09744-f004:**
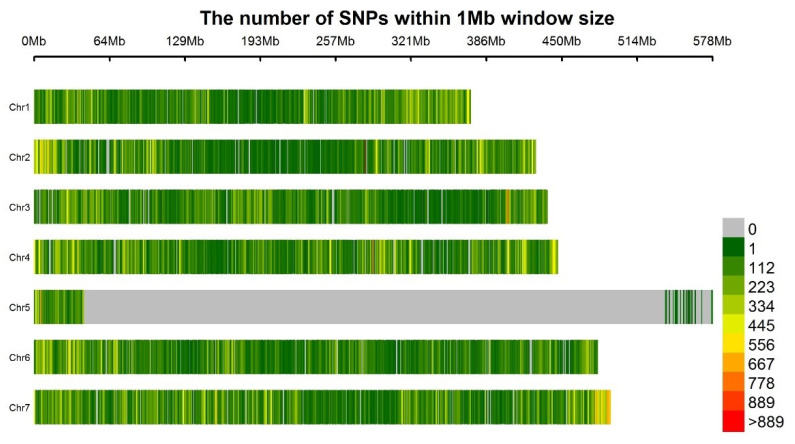
Distribution of polymorphic SNPs differing between Aphanomyces root rot-resistant (R) and susceptible (S) pea in three individual R-S pairs and two in silico mix bulks on the seven pea chromosomes. The colors indicate SNP density (SNPs/Mb) as per the scale on the right-hand side.

**Figure 5 ijms-23-09744-f005:**
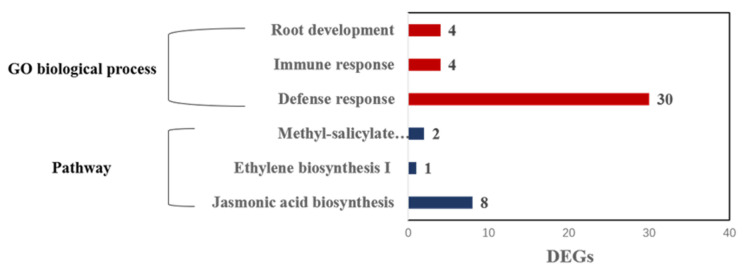
Visualization of number of differentially expressed genes in pea resistant or susceptible to Aphanomyces root rot based on the GO biological process, including root development (GO:0010015, GO:0010053, GO:0022622 and GO:0048364), immune response (GO:0006955) and defense response (GO:0006952, GO:0031347 and GO:0031348), as well as signaling pathways involving salicylate acid, ethylene and jasmonic acid.

**Table 1 ijms-23-09744-t001:** ANOVA for root rot severity, vigor, dry foliar weight and plant height using the pooled data of a recombinant inbred line (RIL) population of pea inoculated with *Aphanomyces euteiches* in three greenhouse experiments.

Source of Variance	df	Mean Square			
		DS	Vigor	Height	DFWT
Genotype (G)	134	95.1 ***	19.1 ***	1303.2 ***	0.069 ***
Repeat	2	1298.7 ***	192.8 ***	5701.6 ***	0.562 ***
G*Repeat	266	22.4 ***	4.2 ***	122.1 ***	0.018 ***
Residuals	3484	5.3	1.0	41.2	0.006
Heritability		0.77	0.74	0.85	0.75

*** indicates the *p*-value that less than 0.001.

## Data Availability

ANOVA table of phenotypic data, RNA sequencing analysis, information of differential expressed genes and detected polymorphic variants between R and S bulks are available in the main manuscript or as Appendix A.

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
