# Peer review of "Identification of Novel Genes Associated with Partial Resistance to Aphanomyces Root Rot in Field Pea by BSR-Seq Analysis"

_ijms, 2022, doi:10.3390/ijms23179744_

Round 1

Reviewer 1 Report

Congratulations on the excellent results and the scientific article.

Aphanomyces euteiches is known to thrive very well in wet and cold conditions. Try adjusting your test settings to lower daytime temperatures and higher substrate humidity.

Aphanomyces euteiches is a serious fungal disease of field and garden pea plants.

It manifests itself mainly in cold and wet seasons and causes serious economic losses. The polygenic inheritance of tolerance to Aphanonyces root rot complicates the most effective genetically based protection.

The submitted manuscript contributes to the development of SNPs for detecting novel QTL controlling partial resistance to A. euteiches in field pea. They indicate the pea cultivar '00-2067' which can be used in resistance breeding programs and for the development of markers for use in MAS.

It is clearly written and meets all the requirements for publication in a journal.

I recommend its publication in present form.

Author Response

Dear Reviewer 1,

Thanks for your positive assessment on our paper. Please see our reply to your comment  in the attachment.

Sincerely,

Longfei Wu

Congratulations on the excellent results and the scientific article.

REPLY: Thank you for the positive assessment.

Aphanomyces euteiches is known to thrive very well in wet and cold conditions. Try adjusting your test settings to lower daytime temperatures and higher substrate humidity.

REPLY: Thank you for your suggestion. We were able to get good disease pressure under our conditions, although we will try lower daytime temperatures in the future.

Aphanomyces euteiches is a serious fungal disease of field and garden pea plants.

REPLY: Indeed, the pathogen can be a serious issue in pea plants, and that is why we are interested in methods to manage this disease.

It manifests itself mainly in cold and wet seasons and causes serious economic losses. The polygenic inheritance of tolerance to Aphanonyces root rot complicates the most effective genetically based protection.

REPLY: This is true, and we touch upon some of the importance of this disease in our introduction and discussion sections as appropriate.

The submitted manuscript contributes to the development of SNPs for detecting novel QTL controlling partial resistance to A. euteiches in field pea. They indicate the pea cultivar '00-2067' which can be used in resistance breeding programs and for the development of markers for use in MAS.

It is clearly written and meets all the requirements for publication in a journal.

I recommend its publication in present form.

REPLY: Thank you for your recommendation. We are happy that you were pleased with the paper.

Reviewer 2 Report

The manuscript is scientifically sound and touches the important topic of disease resistance of a crop plant. Pisum sativum is a well-known and worldwide grown pulse crop, the source of plant protein of the very high value. Peas is also a very good source of macro and microelements, including zinc and some vitamins important for human and animal health, such as A, B, C and E. The yielding of peas is sometimes very low, due to numerous adverse abiotic and biotic conditions, as well as the shape of the plant itself. Its stem is usually not sturdy enough for a higher number of pods with seeds/cannot lift the high yield. Moreover the plant can be damaged by numerous above-ground and soil-borne pathogens, including viruses, bacteria, fungi and oomycetes. Among them, Aphanomyces euteiches is a serious soil-borne pathogen, difficult to control, due to many reasons. Therefore the aim of this study is ambitious and the results are worth publishing. Generally the manuscript is ready to publish, but some additions or explanations could improve its understanding.

Paragraph 2 of the introduction should say if there is any variation in A. euteiches worldwide. Is the experiment done in Canada of worldwide importance and reproducible using isolates from Europe or Asia? Is the isolate Ae-MRDC1 a typical isolate of this oomycete pathogen? Are there any peculiarities in the pathogen populations?

The assessment scale was 0-9. The score >5.5 was called susceptible. Have you found intermediate and fully susceptible reactions? Were there any important differences between these two sub-groups? Can you describe in more detail the R and S disease symptoms observed?

I suggest to add the explanation how the identification of novel genes associated with partial resistance to Aphanomyces root rot in field pea by BSR-seq may facilitate efforts to improve management of this important disease. Resistance to Aphanomyces is mainly based on QTLs, hard for breeding but partially manageable. How the knowledge and identification of 31 genes of general pathways for resistance can be improved in practice, if not by breeding and resistance tests? Can the Authors add a comment to the Discussion?

Please thoroughly check the manuscript for minor mistakes:

·        second author has no affiliation,

·        a few places with double spacing

·        Table 1 please use italics in Latin name of the pathogen

·        reference 19: sativum (not Sativum) and probably some others in the literature list.

I suggest to accept the manuscript after a minor revision and the answer to above-mentioned comments.

Author Response

Dear Reviewer 2,

We really appreciate your positive assessment and valuable suggestions, which improved the manuscript very much. I have replied all your comments and modified the manuscript following your suggestion point by point. Please see the attachment.

Thank you very much.

Sincerely,

Longfei Wu  

The manuscript is scientifically sound and touches the important topic of disease resistance of a crop plant. Pisum sativum is a well-known and worldwide grown pulse crop, the source of plant protein of the very high value. Peas is also a very good source of macro and microelements, including zinc and some vitamins important for human and animal health, such as A, B, C and E. The yielding of peas is sometimes very low, due to numerous adverse abiotic and biotic conditions, as well as the shape of the plant itself. Its stem is usually not sturdy enough for a higher number of pods with seeds/cannot lift the high yield. Moreover the plant can be damaged by numerous above-ground and soil-borne pathogens, including viruses, bacteria, fungi and oomycetes. Among them, Aphanomyces euteiches is a serious soil-borne pathogen, difficult to control, due to many reasons. Therefore the aim of this study is ambitious and the results are worth publishing. Generally the manuscript is ready to publish, but some additions or explanations could improve its understanding.

REPLY: Thank you for the positive assessment of our manuscript.

Paragraph 2 of the introduction should say if there is any variation in A. euteiches worldwide. Is the experiment done in Canada of worldwide importance and reproducible using isolates from Europe or Asia? Is the isolate Ae-MRDC1 a typical isolate of this oomycete pathogen? Are there any peculiarities in the pathogen populations?

REPLY: Thank for your questions regarding the inoculum used. The isolate Ae-MRDC1 was classified as pathotype I (this is now indicated in the materials and methods), which is the most prevalent and virulent pathotype worldwide. We have included some information on the importance of this pathotype in paragraph 2 of the introduction; it is an important pathotype worldwide.

The assessment scale was 0-9. The score >5.5 was called susceptible. Have you found intermediate and fully susceptible reactions? Were there any important differences between these two sub-groups? Can you describe in more detail the R and S disease symptoms observed?

REPLY: Thank you for this suggestion. We have added a brief description of the disease rating scale to the materials and methods, as well as the reason that we regarded a score >5.5 as susceptible.

I suggest to add the explanation how the identification of novel genes associated with partial resistance to Aphanomyces root rot in field pea by BSR-seq may facilitate efforts to improve management of this important disease. Resistance to Aphanomyces is mainly based on QTLs, hard for breeding but partially manageable. How the knowledge and identification of 31 genes of general pathways for resistance can be improved in practice, if not by breeding and resistance tests? Can the Authors add a comment to the Discussion?

REPLY: We have expanded the discussion to address your suggestion, addressing how the identification of novel genes may facilitate efforts to improve management of this disease, particularly with respect to QTL analysis and RNA-seq and their application to breeding programs. Please refer to the seventh and eight paragraphs of the revised discussion.

Please thoroughly check the manuscript for minor mistakes:

REPLY: Thank you – we have carefully reviewed the manuscript and incorporated minor editorial revisions as needed.

  • second author has no affiliation,

REPLY: Corrected

  • a few places with double spacing

REPLY: All of the double spaces were checked and corrected

  • Table 1 please use italics in Latin name of the pathogen

REPLY: Corrected

  • reference 19: sativum (not Sativum) and probably some others in the literature list.

REPLY: The errors have been corrected in reference 19 as well as the rest of manuscript, including the list of references

I suggest to accept the manuscript after a minor revision and the answer to above-mentioned comments.

REPLY: Thank you for the recommendation